# Optimal Wavelengths for Multispectral Short Wavelength Infrared Transillumination and Reflectance Imaging for Caries Detection

**DOI:** 10.3390/diagnostics15081034

**Published:** 2025-04-18

**Authors:** Daniel Fried, Yihua Zhu

**Affiliations:** Department of Preventive and Restorative Dental Sciences, University of California, San Francisco, 707 Parnassus Ave, San Francisco, CA 94143, USA; yihua.zhu@ucsf.edu

**Keywords:** SWIR imaging, dental caries, reflectance imaging, transillumination

## Abstract

**Background/Objectives:** The aim of this in vitro study was to determine the optimal combinations of wavelengths for short wavelength infrared (SWIR) multispectral transillumination and reflectance imaging of caries lesions on proximal and occlusal surfaces. **Methods:** The contrasts of (*n* = 76) caries lesions on the occlusal and proximal surfaces of extracted teeth were measured at 1050, 1300, and 1550 nm for occlusal transillumination and 1058, 1300, 1450, and 1675 nm for occlusal reflectance. All teeth were also imaged using radiography and microcomputed tomography (μCT) to verify lesion presence. A custom-fabricated handheld imaging probe suitable for clinical use and for the simultaneous acquisition of SWIR occlusal transillumination and reflectance (SWIR-OTR) images was used. Three high-power superluminescent diode lasers were used for transillumination, and a fiber-optic switch was used to switch between the transillumination wavelengths. Optical bandpass filters coupled with a tungsten halogen lamp were used for reflectance. All images were acquired at the same position and with the same field of view for comparison. **Results:** The highest contrasts in reflection were at 1450 and 1675 nm for occlusal and interproximal lesions, and the highest contrasts for transillumination were at 1050 and 1300 nm. **Conclusions:** This study suggests that the best wavelengths for SWIR-OTR are between 1000 and 1300 nm for transillumination and greater than 1400 nm for reflectance. Wavelengths beyond 1400 nm are advantageous for reflectance and yield significantly higher contrast. Wavelengths beyond 1300 nm are not promising for occlusal transillumination since internal water absorption leads to contrast inversion.

## 1. Introduction

Despite the introduction of fluoridated drinking water, fluoride dentifrices and rinses, and improved dental hygiene in the US, dental decay continues to be the leading cause of tooth loss [1]. The locations of most newly discovered caries are on the occlusal pits/fissures of the posterior dentition and the interproximal contact sites between adjacent teeth, where radiographs and visual diagnosis have low diagnostic sensitivity due to overlapping enamel and difficulty of physical access [2,3]. New diagnostic tools are needed that do not require the use of ionizing radiation. The risk of exposure to ionizing radiation is poorly understood and even greatly reduced levels of radiation exposure may still pose a significant risk especially for children and pregnant women.

Short wavelength infrared (SWIR) imaging is highly promising for imaging dental caries due to the high transparency of dental enamel at these longer wavelengths and the lack of interference from stains [4]. SWIR and near-IR imaging (NIR) methods have been under development for more than 20 years for use in dentistry, and several NIR clinical devices are now available [4]. The first clinical SWIR imaging study was carried out in 2010, using proximal (SWIR-PT) and occlusal transillumination (SWIR-OT) devices operating at 1300 nm [5]. Currently, the only clinical systems that are available operate at shorter NIR wavelengths at 830 and 780 nm [6,7,8,9]. The lower light scattering of enamel results in very high contrast of caries lesions at SWIR wavelengths. The magnitude of that contrast greatly exceeds radiography and is more sensitive to mineral loss than radiographic methods. Lesions are not typically visible on radiographs until decalcification has exceeded 30% [3,10], while SWIR imaging methods are diagnostic with only 5% decalcification [4]. The high contrast at SWIR wavelengths is advantageous for the implementation of automated caries-detection algorithms that are challenging for radiography due to the very low lesion contrast. SWIR imaging is particularly well suited for use with artificial intelligence (AI). Casalegno showed that AI could be used to analyze clinical NIR transillumination images at 780 nm [11].

SWIR and NIR imaging methods have been shown to have much higher sensitivity than radiographs for the detection of lesions on both occlusal and proximal tooth surfaces [12,13,14]. One potential problem with systems with higher sensitivity is the increasing possibility of false positives that may lead to overtreatment. Radiography suffers from low sensitivity particularly for occlusal lesions; however, it does have very high specificity. Imaging at wavelengths beyond 1200 nm can reduce the possibility of false positives due to stains; however, cracks, hypomineralization, and other structural features in teeth can appear similar to caries lesions and produce false positives.

Multiple approaches have been used to image tooth surfaces using SWIR imaging [12]. For SWIR proximal transillumination (SWIR-PT), the light source and detector are placed on opposite sides of the proximal contact [12]. Two views can be captured for each contact with the detector being placed on either the buccal or the lingual side of the tooth. This method is only effective for detecting proximal lesions. For SWIR occlusal transillumination (SWIR-OT), light is delivered using two fiber-optic conduits positioned low on the tooth beneath lesion areas, and the tooth is imaged from the occlusal surface [4]. For SWIR occlusal reflectance (SWIR-R), the light source and detector are both placed above the occlusal surface. Cross-polarization is required to reduce specular reflection from tooth occlusal surfaces that can be quite strong due to the high refractive index of dental enamel. Both configurations can be used to detect lesions on occlusal and proximal surfaces, and they can be easily combined into a single occlusal transillumination and reflectance (SWIR-OTR) imaging probe [13,15,16]. Transillumination and reflectance images can be acquired near simultaneously at multiple wavelengths using the SWIR-OTR probe and high-speed fiber-optic switches to increase specificity and decrease the number of false positives [13,15,16]. Even though two clinical studies have been successfully carried out using the SWIR-OTR probe [13,14], the optimum wavelengths to employ for SWIR-OTR have not yet been established.

Clinicians assume that lesions that appear on radiographs are deeper than they appear because radiographs significantly underestimate the lesion depth [17,18,19]. Multispectral SWIR transillumination and reflectance measurements can also be used to provide estimates of lesion severity. Studies suggest that transillumination performs best at 1300 nm where the transparency of enamel is high and water absorption is still low while the contrast for reflectance is highest at 1950 nm [4]. Zakian et al. [20,21] used SWIR hyperspectral reflectance imaging to show that teeth appear darker at longer SWIR wavelengths and to estimate the severity of occlusal lesions. Zhu et al. [15] used simultaneous SWIR-OT (1300 nm) and SWIR-R (1610 nm) images to estimate the severity of occlusal and interproximal lesions on extracted teeth by fusing the images together.

Another reason for using multiple wavelengths for imaging is that at wavelengths beyond 1400 nm with higher water absorption and lower light scattering, teeth appear very dark particularly when wet. Tooth surface tomography can be better viewed at 1050 nm even though the contrast between sound and lesion areas is much higher at 1450 nm. By adding a second 1050 nm wavelength of low water absorption and higher scattering in sound enamel and dentin, tooth surfaces are more clearly visible when wet.

Simon et al. [22] assembled a desktop multispectral SWIR imaging system that was capable of acquiring both transillumination and reflectance images of occlusal and proximal lesions at wavelengths from 1200 to 1700 nm. The system used a high-power tungsten halogen lamp and filters to acquire images at each wavelength. Transillumination imaging requires much higher intensities than reflectance, and the transillumination wavelength bands utilized were broad and the overall intensities were low; 1200–1700 and 1500–1700 nm wavelength bands were used, and it was not possible to show whether it is advantageous to use wavelengths longer than 1300 nm for occlusal transillumination. The system used in this study utilizes high-power superluminescent diodes centered at 1050, 1300, and 1550 nm for transillumination to achieve higher intensities than those used in prior studies. Three wavelengths were employed for SWIR-OT, 1050, 1300, and 1550 nm, and four wavelengths were used for reflectance, 1058, 1300, 1450, and 1675 nm. Images were all acquired using a SWIR-OTR handpiece so that all the acquired images were acquired at the same position and field of view. The purpose of this study was to measure and compare the contrast of occlusal and interproximal lesions at these seven wavelengths to establish the optimal wavelengths to be used for SWIR-OTR.

## 2. Materials and Methods

### 2.1. Sample Preparation Amd Microcomputed Tomography (µCT)

Teeth (*n* = 74) with no identifiers were collected from patients in the San Francisco Bay area with (*n* = 76) occlusal and interproximal lesions for this study. Such collection is exempt from IRB approval, Exemption 45 CFR 46.104 Category 4 from the U.S. Department of Health and Human Services (HHS) and is not considered human subject research. Teeth were collected (1–5 years) prior to imaging, sterilized using gamma radiation, refrigerated, and stored with a 0.1% thymol solution to maintain tissue hydration and prevent bacterial growth. Lesions were selected based on their presence on µCT images and visibility in at least one of the SWIR images. Lesions were chosen that had penetrated at least half-way through the enamel. Large cavitated lesions or those that penetrated more than half-way through dentin were avoided. Samples were mounted in black orthodontic acrylic blocks from Great Lakes Orthodontics (Tonawanda, NY, USA) and imaged with digital radiographs using a CareStream 2200 System from Kodak (Rochester, NY, USA) operating at 60 kV.

All teeth were imaged using Microcomputed X-ray tomography (µCT) with a 10 µm resolution. A Scanco µCT 50 from Scanco USA (Wayne, PA, USA) was used to acquire the images.

### 2.2. Multispectral Image Acquisition and Analysis

Visible color images of the samples were acquired using a USB microscope, Model AM7915MZT from AnMO Electronics Corp. (New Taipei City, Taiwan) with an extended depth of field and cross-polarization. The digital microscope captures 5-mega-pixel (2952 × 1944) color images.

The SWIR reflectance images were captured using a Model GA1280J (Sensors Unlimited, Princeton, NJ, USA) imaging array with a 1280 × 1024-pixel format, a 15 μm pixel pitch, and a bit depth of 12 bit. Details regarding the fabrication of the dual occlusal transillumination and reflectance SWIR-OTR handpiece have been described previously [15]. A diagram and image of the system is shown in Figure 1. Two 25.4 mm diameter planoconvex antireflection coated lenses of 125 mm and 100 mm focal length along with an adjustable aperture were placed between the handpiece and the InGaAs camera to provide a field of view of 11 mm × 11 mm at the focal plane. A low-OH optical fiber of 1 mm diameter was used to deliver light from a stabilized tungsten IR light source, Model SLS202, from Thorlabs (Newton, NJ, USA), with a peak output at 1500 nm. Four bandpass filters, with wavelengths (bandwidth) centered at 1058(78), 1300(90), 1460(85), and 1675(90) nm, were used. The four filters can be switched without changing the position of the sample. The light for reflectance passes through a polarizing beam-splitting cube before incident on the tooth, and a linear polarizer was placed before the camera to achieve cross-polarization for glare reduction. The intensity delivered to the tooth was 5–10 mW. The transillumination light was delivered through two 0.4 mm diameter low-OH optical fibers. Three light sources were used for transillumination: Thorlabs Model SLD 1550S-A40 operating at 1555 nm with a bandwidth of 45 nm and a peak output power of 39 mW, Thorlabs Model SLD 1050S-A60 with a bandwidth of 83 nm and a peak output of 72 mW, and a Model DL-CS3452A-FP 1620-2111 SLD from Denselight (Singapore) operating at 1314 nm with a bandwidth of 33 nm and a peak output of 48 mW. The three light sources were connected to a Model FTS-FOSW 1 × 4 fiber-optic switch connected to a Newport (Irvine, CA, USA) Model 8200 Photonics Test System so that each of the three wavelengths could be switched without changing the position of the sample. A 50/50 beamsplitter was used to deliver light to each arm for transillumination. The output intensity of each arm was approximately 10 mW before entering the Teflon plugs.

The samples were dried of excess water with an air nozzle before imaging due to the strong water absorption at wavelengths beyond 1400 nm [4]. Intensities were measured from a small area within each interproximal lesion that was within the outer enamel. Intensities were chosen to represent sound areas from areas outside the lesion on both sides of the lesion, also in the outer enamel, and the two areas were averaged. For occlusal lesions, only the intensities of sound and lesion areas over underlying dentin were chosen for analysis. Lesion areas were confirmed using the μCT images. Image processing of the images was performed by custom scripts written using MATLAB R2024a from Mathworks (Natick, MA, USA). The acquired 12-bit images (4096) were converted to 16-bit (65,536) by multiplying by 16 and subtracting 1 to facilitate processing using MATLAB.

The mean contrast was calculated for each lesion using three approaches. For the traditional approach, the formula (I_L_ − I_S_)/I_L_ was used for reflectance images, and (I_S_ − I_L_)/I_S_ was used for transillumination images, where I_L_ is the mean intensity of the lesion, and I_S_ is the mean intensity of the sound area [5]. Many of the measurements, particularly those at 1550 nm, yielded negative contrast; therefore, two other methods were used to calculate the contrast. For the second approach, the absolute (ABS) value approach, both reflectance and transillumination areas were calculated using ABS[(I_L_ − I_S_)/I_L_]. In the third approach, the High/Low approach (HL), the formula (I_H_ − I_LW_)/I_H_ was used, where I_H_ is the mean intensity of the area with the highest intensity whether it is a sound or lesion area, and I_LW_ is the mean intensity of the area of lower intensity.

A repeated Measures ANOVA with a Tukey’s multiple comparison test was used to compare the lesion contrast across wavelengths for each lesion type and contrast calculation method. Prism 6 statistical software from GraphPad Software, Inc. (La Jolla, CA, USA) was used for the calculations. Significance level was set at *p* < 0.05.

## 3. Results

The seven SWIR occlusal reflectance and transillumination images are shown in Figure 2 for a tooth with an interproximal lesion that is not visible on the radiograph. The μCT image shows that it is a large lesion that penetrates into dentin. The lesion is clearly visible at all seven of the SWIR wavelengths with high contrast. It is not visible in the color image (VIS), i.e., at 400–700 nm. The contrast is highest for reflectance at 1450 nm where it is 0.81 and lowest at 1058 nm where it is 0.34.

A second tooth with occlusal lesions is shown in Figure 3. The occlusal lesion area designated by the red box appears with extremely high positive contrast in all the SWIR reflectance images. The highest contrast is at 1450 nm where it is 0.95. The lesion is clearly visible with positive contrasts at 1050 and 1300 nm, and it is visible at 1550 nm; however, the occlusal lesion areas now appear with negative contrast at this wavelength for occlusal transillumination, i.e., the contrast has become inverted. A small interproximal lesion also appears in the μCT image, but it is not visible in any of the other images.

A third tooth with both occlusal and interproximal lesions is shown in Figure 4. The interproximal lesion can be seen in the μCT image and all the SWIR images; however, it is not visible in the color image or the radiograph. The interproximal lesion is visible in all the SWIR reflectance images with high contrasts of 0.43, 0.32, 0.58, and 0.56 at 1058, 1300, 1450, and 1675 nm, respectively. However, the contrasts were 0.26, 0.22, and −0.03 for transillumination at 1050, 1300, and 1550 nm. The contrast becomes inverted at 1550 nm, and the lesion area appears brighter than the surrounding sound area for transillumination. The same phenomenon occurs for the occlusal decay in the fissure. The occlusal decay in the fissure is visible with high contrast in the 1450 and 1675 nm images with contrasts of 0.79 and 0.67, respectively, at the position indicated by the yellow box. The contrasts are much lower at 1058 and 1300 nm with values of 0.18 and 0.2, respectively. The contrasts for transillumination were 0.15, 0.10, and −0.28 at 1050, 1300, and 1550 nm. Again, the contrast is inverted for transillumination at 1550 nm.

Contrast inversion is most prevalent at 1550 nm for transillumination. The longer the wavelength, the more likely the lesion contrast will switch from positive to negative. Out of the twenty proximal lesions, the contrast for occlusal transillumination at 1550 nm was negative for four of the lesions while it was positive for all the lesions at 1300 and 1050 nm. The contrast was positive for all the interproximal lesions for occlusal reflectance measurements at all four wavelengths. For the 56 occlusal lesions, the contrast for occlusal transillumination was negative for 6 of the lesions at 1050 nm, 12 of the lesions at 1300 nm, and 35 of the lesions at 1550 nm.

Three of the occlusal lesions appeared with negative contrast at 1058 nm for occlusal reflectance measurements due to the presence of stains.

The mean and standard deviation of the contrast is tabulated in Table 1 for the *n* = 20 interproximal lesions and the *n* = 56 occlusal lesions. Means are tabulated using all three methods for calculating the contrast: conventional, ABS, and HL. There were no differences in the mean contrast in reflectance for interproximal lesions and only very small differences in the contrast in mean reflection for the occlusal lesions between the different methods used for calculating the contrast. For transillumination, there was only a small difference in contrast for 1550 nm for the interproximal lesions. There were much larger differences observed for the mean contrast for transillumination of the occlusal lesions, particularly for 1550 nm. The mean contrast was negative for occlusal lesions at 1550 nm using the traditional approach.

## 4. Discussion

The purpose of this study was to determine the optimum transillumination and reflectance wavelengths to use for SWIR-OTR. The motivation for combining reflectance and transillumination is to increase diagnostic performance by reducing false positives. Other advantages that have been investigated include the assessment of lesion depth and severity and the differentiation between cavitated and noncavitated interproximal lesions [4]. In two clinical studies, it has been established that the SWIR-OTR probe can be used effectively in vivo. In those studies, 1300 nm was used for transillumination and 1610 nm for reflectance [13,14]. In this study, we investigated two additional transillumination wavelengths of 1050 and 1550 nm and four reflectance wavelengths.

Reflectance at 1058 nm and 1300 nm was also included because it is also desirable to better visualize the surface of the tooth, and at longer wavelengths beyond 1400 nm the surface of sound areas of the tooth may not be visible due to the lower scattering of sound enamel and the increase in water absorption [4]. This is particularly challenging when surfaces are wet. Time-resolved SWIR reflectance imaging during the drying of lesions with forced air can be used to assess the activity of lesions [4], and it may be difficult to view the lesions to optimally position the imaging probe directly over the lesions being investigated. The use of a second reflectance wavelength may greatly aid alignment. In addition, image registration of the sequential images may benefit from the higher reflectivity, particularly at 1058 nm. The reflectance images shown in Figure 2, Figure 3 and Figure 4 clearly show the increased visibility of sound areas of the tooth compared to 1450 and 1675 nm.

Previous studies involving SWIR transillumination imaging have been mostly limited to 1300 nm. There are multiple reasons for the focus on 1300 nm. It is beyond 1200 nm where absorption due to stains is minimal, and it is below 1400 nm where water absorption increases markedly. Therefore, the enamel has maximum transparency. The further reduction in light scattering in enamel at wavelengths beyond 1300 nm suggests that longer wavelengths such as 1550 nm may be advantageous even though there is a reduction in transparency due to increased water absorption. However, this study shows that contrast inversion may occur at longer SWIR wavelengths such as 1550 nm.

This study sheds light on the mechanism of contrast inversion for occlusal transillumination. We have observed that the contrast may become inverted if the transillumination probe is too high on the tooth and more light is incident above the lesion than below it. This phenomenon has been observed for both in vitro and in vivo imaging of lesions on proximal and occlusal surfaces [4]. This study shows that there is a significant wavelength dependence of this phenomenon in addition to the dependence on the position of the illumination source and the distance of the lesion from the occlusal surface. The longer the wavelength, the more likely the lesion contrast will switch from positive to negative. This study shows that occlusal transillumination (SWIR-OT) is highly dependent on the light paths that the light injected into the tooth follow before leaving the occlusal surface. Ideally, the light will penetrate the interior dentin of the tooth and diffuse upwards through the center of the tooth before exiting the occlusal surface. The light can also remain in the more transparent outer enamel and migrate around the dentin where it is less likely to be absorbed. To successfully acquire high contrast of lesions, particularly occlusal lesions, the number of photons that penetrate the dentin and pass through the lesion and the adjacent sound enamel or dentin from below must exceed the photons that migrate around the enamel and impinge on the lesion and the adjacent sound enamel or dentin from above. Otherwise, the contrast of the lesion will be inverted and appear whiter than the surrounding sound tissues rather than darker, which is necessary for effective SWIR-OT. This phenomenon resulted in a negative mean contrast for the occlusal lesions (−0.22) at 1550 nm while the contrast was positive for 1050 and 1300 nm. Calculation of the contrast using the ABS and HL approaches resulted in positive contrasts of 0.40 and 0.26, respectively. The 0.40 contrast is very high, and at first glance it appears highly promising; however, it is important to note that it truly represents reflective measurements rather than transillumination since most of the photons go around the outside of the tooth and impinge on the lesion from above while most of the photons that enter the tooth in the dentin below the lesion are absorbed by the increased water absorption at 1550 nm.

Reflectance at 1450 and 1675 nm did yield the highest mean contrast exceeding 0.6. Negative contrast was also measured for a few of the occlusal lesions in reflectance at 1058 nm. Negative contrast in reflectance occurs due to absorption by stains. In a previous study measuring the contrast of occlusal lesions at wavelengths from 400 to 2300 nm, the contrast was negative in the visible and 830 nm due to absorption by stains, and stains contributed to absorption at wavelengths less than 1200 nm [23]. Stains can lead to false positives. Even with the increased interference from the absorption by stains that was observed at 1050 and 1058 nm, it was not as significant as expected. This suggests that a wavelength near 1000 nm may effectively replace 1300 nm for transillumination. It may however slightly increase the false positive rate, which is a great concern due to the much higher sensitivity of SWIR imaging compared to radiography.

This study used a larger imaging array, the same that was used in an earlier clinical study [12], which is larger than the smallest InGaAs imaging arrays that are currently available. The smaller arrays were used for our more recent clinical studies using SWIR-OTR [13,14]. The 3D printed OTR probe was identical to OTR probes that were used for those two SWIR-OTR [13,14] clinical studies, and the results of this study should apply equally to the SWIR-OTR systems using the smaller InGaAs imaging array.

No dental SWIR imaging systems are currently available in the commercial market. Current NIR imaging devices on the market for caries detection operate at 780 and 850 nm. Lesion contrast is lower at these wavelengths, and stains absorb the NIR light [23]. The primary disadvantage of operating at longer SWIR wavelengths is that Si-based imaging technologies are only efficient at wavelengths under 1000 nm. High costs for SWIR InGaAs imaging arrays have been mainly due to limited production and restrictions on the use of technology with military applications. Prices are dropping rapidly with increased competition and the introduction of many nonmilitary applications including medical thermography, firefighting, enhanced vision, security/surveillance, and machine vision. High-resolution thermal imaging devices that were once prohibitively expensive are now relatively inexpensive. New SWIR imaging technologies such as quantum dots are under development, and they offer higher performance at lower cost over InGaAs for large imaging arrays and have improved sensitivity at SWIR wavelengths beyond 1700 nm [24].

This in vitro study lacks the presence of adjacent teeth and the conditions of the oral environment that would be encountered for clinical imaging. Those factors more profoundly influence diagnostic performance comparisons with radiography and should be less important for the lesion contrast comparisons between the SWIR wavelengths that were carried out in this study.

## 5. Conclusions

This study suggests that the best wavelengths for SWIR-OTR are between 1000 and 1300 nm for transillumination and greater than 1400 nm for reflectance. Wavelengths beyond 1450 nm are advantageous for occlusal reflectance and yield significantly higher contrast. Occlusal transillumination is more complex; it may benefit from use of a shorter wavelength near 1050 nm that yields similar contrast to 1300 nm and has better visibility of sound tooth surfaces with only minor interference from stains. The use of longer wavelengths beyond 1300 nm reduces the effectiveness of occlusal transillumination due to increased contrast inversion from higher water absorption.

## Figures and Tables

**Figure 1 diagnostics-15-01034-f001:**
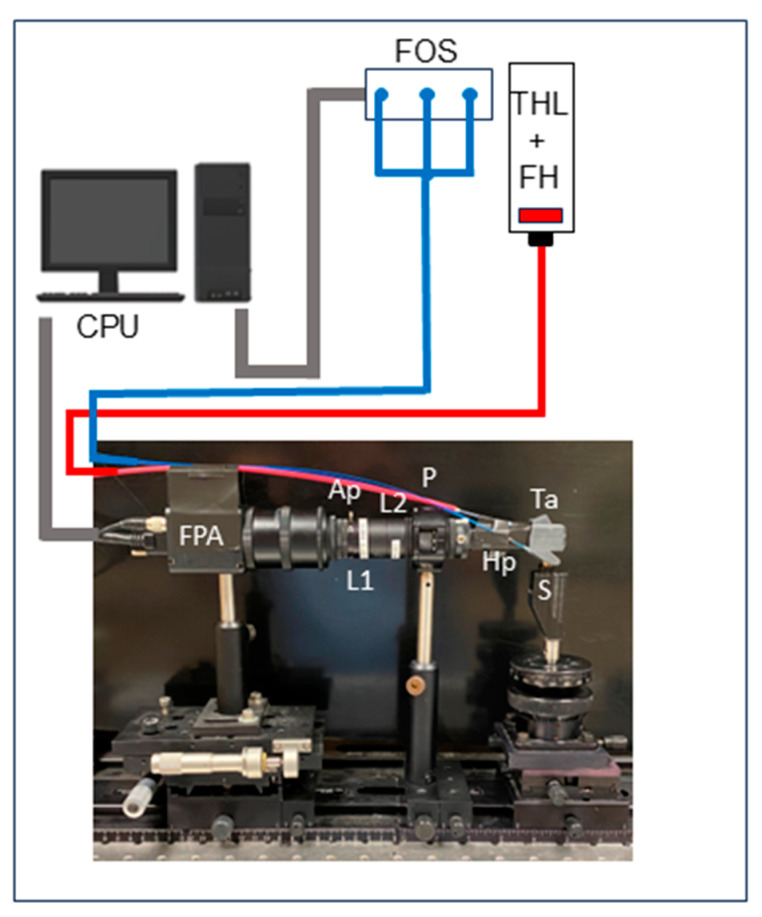
Imaging system with a computer (CPU); three SLDs with fiber-optic switches (FOSs) are used for transillumination, and a tungsten halogen lamp (THL) with a filter holder (FH) with bandpass filters is used for reflectance. An InGaAs focal plane imaging array (FPA) with external optics and handpiece is positioned over the tooth (S). The red fiber is for reflectance, and the blue fibers are for transillumination. The system includes a 100 mm lens (L1), polarizer (P), 60 mm lens (L2), adjustable aperture (Ap), reflectance handpiece (Hp), and transillumination attachment (Ta).

**Figure 2 diagnostics-15-01034-f002:**
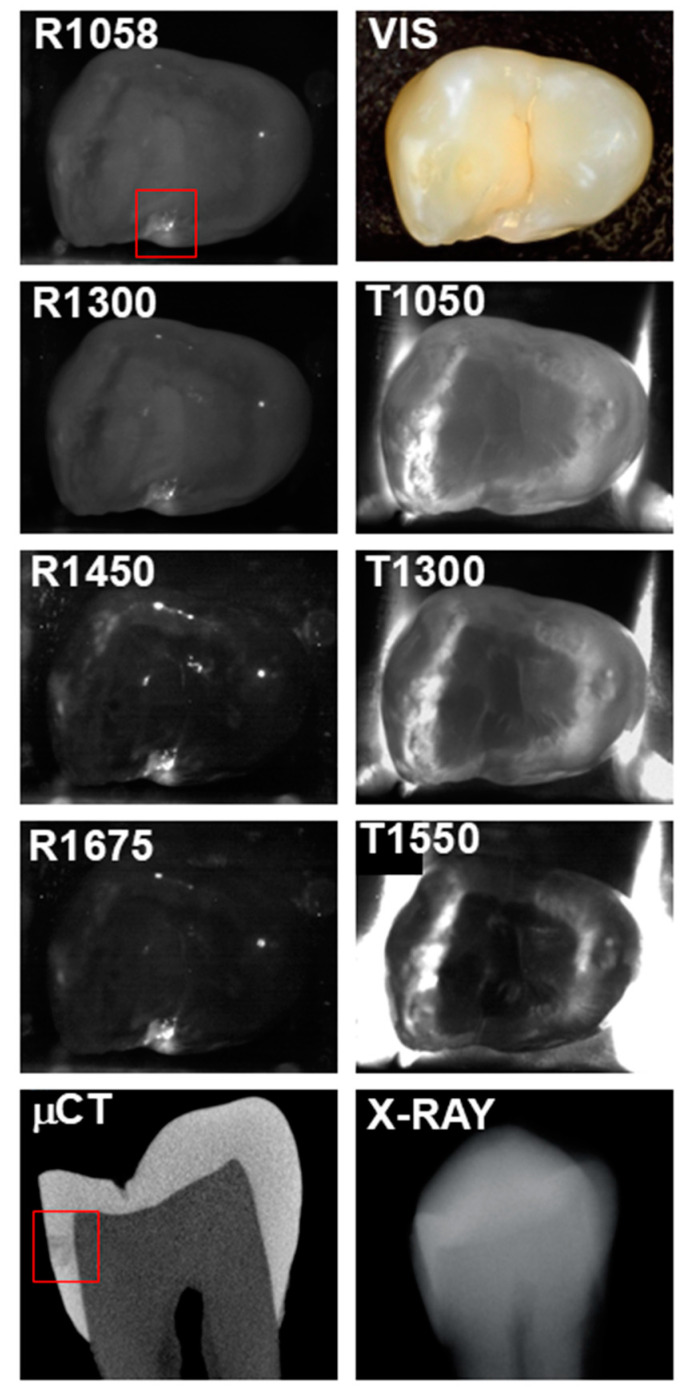
SWIR occlusal reflectance (R) images at 1058, 1300, 1450, and 1675 nm along with SWIR occlusal transillumination (T) images at 1050, 1300, and 1550 nm are shown for a tooth with an occlusal lesion enclosed in the red box. A color (VIS) image, a slice extracted from the μCT image, and a radiograph (X-RAY) are also shown.

**Figure 3 diagnostics-15-01034-f003:**
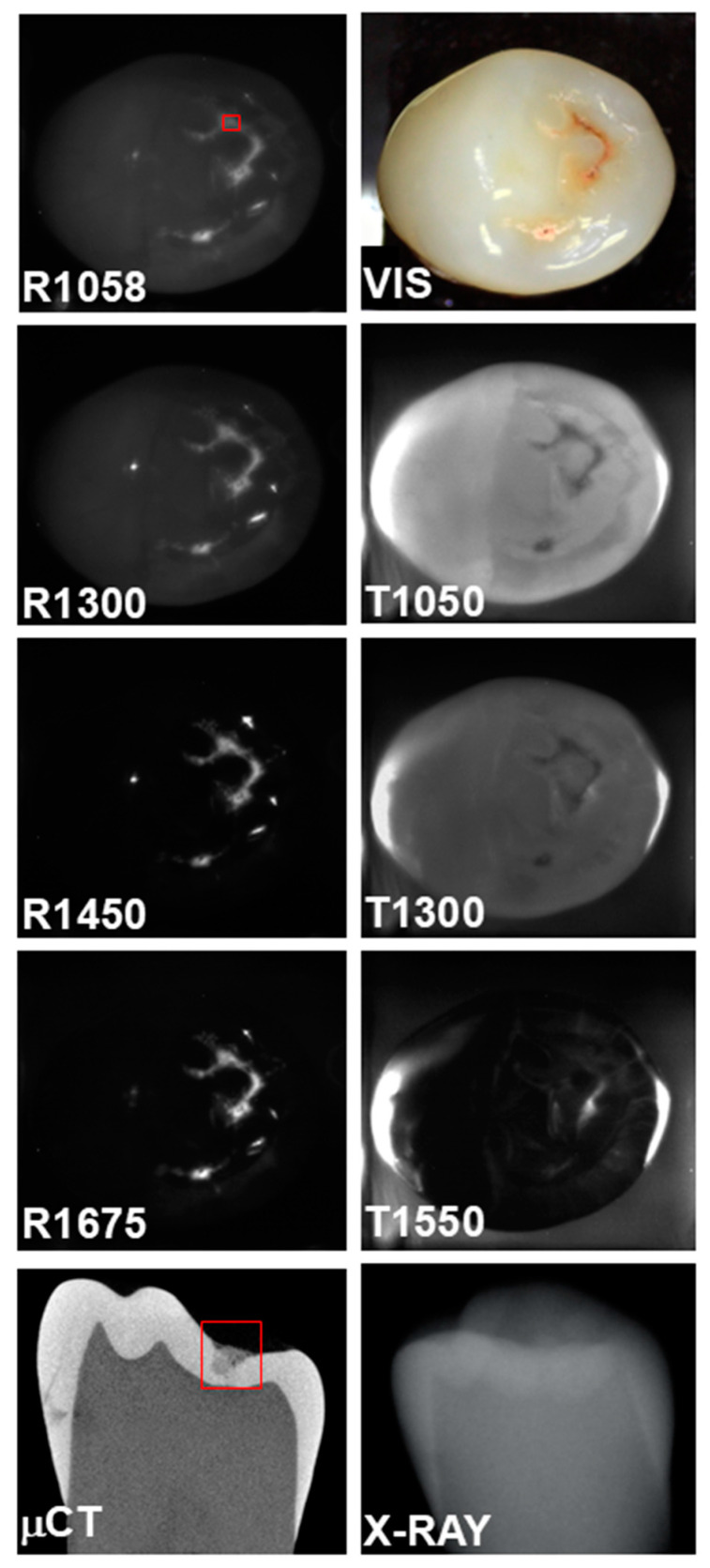
SWIR occlusal reflectance (R) images at 1058, 1300, 1450, and 1675 nm along with SWIR occlusal transillumination (T) images at 1050, 1300, and 1550 nm are shown for a tooth with an occlusal lesion enclosed in the red box. A color (VIS) image, a slice extracted from the μCT image and a radiograph (X-RAY) are also shown.

**Figure 4 diagnostics-15-01034-f004:**
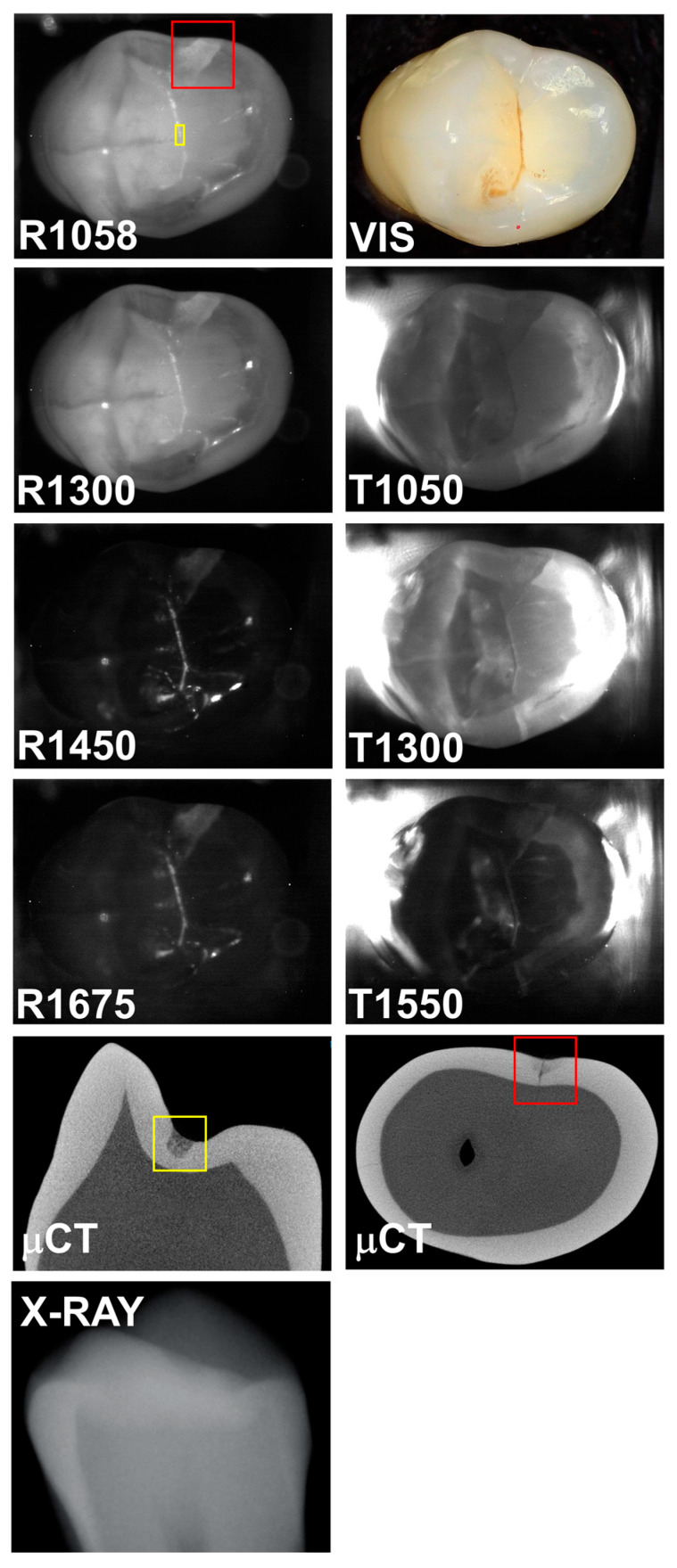
SWIR occlusal reflectance (R) images at 1058, 1300, 1450, and 1675 nm along with SWIR occlusal transillumination (T) images at 1050, 1300, and 1550 nm are shown for a tooth with an occlusal lesion (yellow box) and an interproximal lesion (red box). A color (VIS) image, two slices extracted from the μCT image, and a radiograph (X-RAY) are also shown.

**Table 1 diagnostics-15-01034-t001:** Mean lesion contrast ± s.d. for SWIR methods for the *n* = 20 interproximal lesions and *n* = 56 occlusal lesions. Means with the same letter across each row are statistically similar (*p* > 0.05). ABS means were calculated using the absolute values of the contrast, and HL means were calculated using the formula (I_H_ − I_LW_)/I_H_, regardless of whether the lesion intensity was higher than the sound intensity.

Mean Lesion Contrast ± s.d.	Reflectance 1058 nm	1300 nm	1450 nm	1675 nm	Transillumination 1050 nm	1300 nm	1550 nm
Proximal	0.30 ± 0.11 ^b^	0.32 ± 0.11 ^b^	0.53 ± 0.15 ^a^	0.57 ± 0.14 ^a^	0.27 ± 0.10 ^b^	0.24 ± 0.12 ^b^	0.20 ± 0.20 ^b^
Occlusal	0.27 ± 0.17 ^b,c^	0.29 ± 0.15 ^b^	0.65 ± 0.22 ^a^	0.65 ± 0.22 ^a^	0.19 ± 0.15 ^c^	0.12 ± 0.21 ^c^	−0.22 ± 0.62 ^d^
ProximalABS	0.30 ± 0.11 ^b^	0.32 ± 0.11 ^b^	0.53 ± 0.15 ^a^	0.57 ± 0.14 ^a^	0.27 ± 0.10 ^b^	0.24 ± 0.12 ^b^	0.23 ± 0.16 ^b^
OcclusalABS	0.28 ± 0.16 ^b^	0.29 ± 0.14 ^b^	0.70 ± 0.46 ^a^	0.68 ± 0.29 ^a^	0.22 ± 0.11 ^c,d^	0.19 ± 014 ^d^	0.40 ± 0.51 ^b,c^
ProximalHL	0.30 ± 0.11 ^b^	0.32 ± 0.11 ^b^	0.53 ± 0.15 ^a^	0.57 ± 0.14 ^a^	0.27 ± 0.10 ^b^	0.24 ± 0.12 ^b^	0.23 ± 0.16 ^b^
OcclusalHL	0.28 ± 0.16 ^b,c^	0.29 ± 0.15 ^b^	0.65 ± 0.22 ^a^	0.65 ± 0.22 ^a^	0.22 ± 0.11 ^c,d^	0.19 ± 0.13 ^d^	0.26 ± 0.19 ^b,c,d^

## Data Availability

The original contributions presented in this study are included in the article. Further inquiries can be directed to the corresponding author.

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
