# Peer review of "Optimal Wavelengths for Multispectral Short Wavelength Infrared Transillumination and Reflectance Imaging for Caries Detection"

_diagnostics, 2025, doi:10.3390/diagnostics15081034_

Round 1

Reviewer 1 Report

Comments and Suggestions for Authors

Dear Author,

I have reviewed your manuscript and I see that there are some points that need to be answered. My comments on this issue are given below.

The introduction is too long and complicated. Please write the purpose of your study by mentioning the different aspects of your study from the literature. Please simplify the introduction.

What is the strength and effect size of the study? Please state. Explain the statistical methods used in the study.

How long ago the extracted teeth were extracted and how they were stored are parameters that may affect the results of the study. Please mention this period and storage conditions in your study.

Have the extracted teeth been evaluated for the presence of conditions such as cracks or hypoplasia? Such conditions may give misleading results in terms of the presence of caries.

How did you determine the 150 nm change between the wavelengths you investigated in your study (1050, 1300, and 1550 nm) ? Please explain.

The depth or volume of the carious lesion may affect the reflections at different wavelengths. Therefore, an evaluation including the depth of the lesion may have an impact on your study results.

Author Response

Thank you very much for taking the time to review this manuscript. We appreciate your efforts to improve the manuscript.  Please find the detailed responses below and the corresponding revisions/corrections highlighted/in track changes in the re-submitted files.

Reviewer#1

I have reviewed your manuscript and I see that there are some points that need to be answered. My comments on this issue are given below.

  1. The introduction is too long and complicated. Please write the purpose of your study by mentioning the different aspects of your study from the literature. Please simplify the introduction.

Response: The introduction has been edited and streamlined.

  1. What is the strength and effect size of the study? Please state. Explain the statistical methods used in the study.

Response: We apologize for leaving out the statistical approach.  This has been added to methods section.

Repeated Measures ANOVA with Tukey’s multiple comparison test was used to compare the lesion contrast across wavelengths for each lesion type and contrast calculation method.  Prism statistical software from GraphPad Software, Inc., (La Jolla, CA) was used for the calculations. Significance level was set at P < 0.05. The statistical results are presented in Table 1.  Means with the same letter across each row are statistically similar (P>0.05).   

  1. How long ago the extracted teeth were extracted and how they were stored are parameters that may affect the results of the study. Please mention this period and storage conditions in your study.

Have the extracted teeth been evaluated for the presence of conditions such as cracks or hypoplasia? Such conditions may give misleading results in terms of the presence of caries.

Response: Added to text- Teeth were collected (1-5 years) prior to imaging, sterilized using gamma radiation, refrigerated and stored with a 0.1% thymol solution to maintain tissue hydration and prevent bacterial growth.   Lesions were selected based on their presence on microCT images and visibility in at least one of the SWIR images.  Lesions were chosen that had penetrated at least half-way through the enamel.  Large cavitated lesions or those that penetrated more than half-way through dentin were avoided.  Cracks and hypomineralization can indeed and lead to false positives and some of the teeth did have these defects.   All lesions were confirmed with microCT to avoid any misdiagnosis.

  1. How did you determine the 150 nm change between the wavelengths you investigated in your study (1050, 1300, and 1550 nm) ? Please explain.

Wavelengths span the sensitivity of standard InGaAs detectors 1000-1750 nm, individual wavelengths are chosen based on the availability of superluminecent diode lasers and bandpass filters.

  1. The depth or volume of the carious lesion may affect the reflections at different wavelengths. Therefore, an evaluation including the depth of the lesion may have an impact on your study results.

Response: The depth and severity of demineralization do impact the lesion contrast.  Therefore, we used RM-ANOVA to compare the lesion contrast over wavelengths for each lesion. 

Reviewer 2 Report

Comments and Suggestions for Authors

Dear Authors,

Thank You For a pleasure to read Your article.

I have several notes to improve Your article.

Title

Please, do not use here short form of words and point the type of study.

Abstract

Please, add one sentence for actuality.

Materials and methods. Please, add information about statistics criteria.

Please, add probabilities for results.

Conclusion is closer for Results. Please, write here the practical use of Your study.

Keywords: please, check them with MeSH.

Materials and Methods

Please, describe statistics section. Also, please, write the sample size calculation.

Results

Lines 272-279 are appropriate for discussion. Please, replace them.

You have many conclusions without proof in this section; please, write last one or avoid them.

Also, You wrote about mean and SD but did not write about distribution character.

It is not clear the why You had this samples number for different lesions.

Discussion

Please, re-write it: this section requires to compare Your results with other study not repeat results and Your previous research.

Please, write the limitations of Your study.

Conclusion is absent. Please, write it

Sincerely, Reviewer

Author Response

Thank you very much for taking the time to review this manuscript. We appreciate your efforts to improve the manuscript.  Please find the detailed responses below and the corresponding revisions/corrections highlighted/in track changes in the re-submitted files.

Reviewer#2

Title

  1. Please, do not use here short form of words and point the type of study.

 Response:  SWIR is now spelled out in the title

Abstract

  1. Please, add one sentence for actuality.

Response: The abstract has been revised as requested

  1. Materials and methods.

Please, add information about statistics criteria. Please, add probabilities for results. Please, describe statistics section. Also, please, write the sample size calculation.

Response: Response: We apologize for leaving out the statistical approach.  This has been added to methods section. Repeated Measures ANOVA with Tukey’s multiple comparison test was used to compare the lesion contrast across wavelengths for each lesion type and contrast calculation method.  Prism statistical software from GraphPad Software, Inc., (La Jolla, CA) was used for the calculations. Significance level was set at P < 0.05. The statistical results are presented in Table 1.  Means with the same letter across each row are statistically similar (P>0.05).   

No sample size calculations were deemed necessary prior to the study since the study only involved extracted teeth that were already collected and there was no clinical or animal approvals required.  In addition, prior studies have shown that similar sample sizes (n=54 and n=20) were sufficient to show statistical significance.   Moreover, large differences (effect sizes) are required to be clinically meaningful.

  1. Keywords: please, check them with MeSH.

Response: dental caries and transillumination are MeSH terms.  SWIR and reflectance imaging are not but are effective and necessary for searching PubMed.

Results

  1. Lines 272-279 are appropriate for discussion. Please, replace them.

You have many conclusions without proof in this section; please, write last one or avoid them.

Response: Sections discussing the mechanism of contrast inversion have been moved to the discussion.

  1. Also, You wrote about mean and SD but did not write about distribution character.

It is not clear the why You had this samples number for different lesions.

 Response: Most groups were normally distributed and nonparametric statistics were utilized. The lesions were selected from a pool of 120 extracted teeth that had microCT data.  There were n=20 proximal lesions and n=54 occlusal lesions.  Lesions were selected based on their presence on microCT images and visibility in at least one of the SWIR images.  Lesions were chosen that had penetrated at least half-way through the enamel.  Large cavitated lesions or those that penetrated more than half-way through dentin were avoided.

Discussion

  1. Please, re-write it: this section requires to compare Your results with other study not repeat results and Your previous research.

Response: The discussion has been edited as requested.

  1. Please, write the limitations of Your study.

Response: We have added to the discussion: This in vitro study lacks the presence of adjacent teeth and the conditions of the oral environment that would be encountered for clinical imaging.   Those factors more profoundly influence diagnostic performance comparisons with radiography and should be less important for the lesion contrast comparisons between the SWIR wavelengths that were carried out in this study.

  1. Conclusion is absent. Please, write it

Conclusion is closer for Results. Please, write here the practical use of Your study.

Response: The conclusion has been added that includes the practical implications of the study, the best choice of SWIR wavelengths to be used for reflectance and transillumination.

Reviewer 3 Report

Comments and Suggestions for Authors

-Neither the abstract nor the text of the manuscript describes what was measured, how the lesions were measured, what were the units of measurement, what comparisons were made, how the statistical analyses were carried out.

-The title uses the acronym SWIR without specifying what it is!

Comments on the Quality of English Language

-The use of scientific English is acceptable.

Author Response

Thank you very much for taking the time to review this manuscript. We appreciate your efforts to improve the manuscript.  Please find the detailed responses below and the corresponding revisions/corrections highlighted/in track changes in the re-submitted files.

Reviewer#3

  1. Neither the abstract nor the text of the manuscript describes what was measured, how the lesions were measured, what were the units of measurement, what comparisons were made, how the statistical analyses were carried out.

We apologize for leaving out the statistical approach.  This has been added to methods section. Repeated Measures ANOVA with Tukey’s multiple comparison test was used to compare the lesion contrast across wavelengths for each lesion type and contrast calculation method.  Prism statistical software from GraphPad Software, Inc., (La Jolla, CA) was used for the calculations. Significance level was set at P < 0.05. The statistical results are presented in Table 1.  Means with the same letter across each row are statistically similar (P>0.05).  

The paragraph on intensity measurements has been rewritten to give a better idea how the intensity measurements were taken: Intensities were measured from a small area within each interproximal lesion that was within the outer enamel.  Intensities were chosen to represent sound areas from areas outside the lesion on both sides of the lesion, also in the outer enamel, and the two areas were averaged.  For occlusal lesions, only the intensities of sound and lesion areas over underlying dentin were chosen for analysis.   

The lesion contrast was compared.  Lesion contrast was measured and calculated using the methods and equations described in Section 2.2.  The units of measurement are dimensionless.

  1. The title uses the acronym SWIR without specifying what it is.

Response:  SWIR is now spelled out in the title

Reviewer 4 Report

Comments and Suggestions for Authors

Dear authors, i believe your article is fit for publication and only needs a couple of changes:

  1. the introduction could be streamlined as it is far too long.
  2. Why was this sample size selected?
  3. Why were these instruments chosen?
  4. The statistical analysis paragraph is missing
  5. LINE 247: The x-ray should be presented in the image.
  6. AUthors should state what are the limits of their study and where they think they could have improven their research.

Author Response

Reviewer#4

Dear authors, I believe your article is fit for publication and only needs a couple of changes:

  1. The introduction could be streamlined as it is far too long.

Response: The introduction has been edited and streamlined.

  1. Why was this sample size selected?

Response: The lesions were selected from a pool of 120 extracted teeth that had microCT data.  There were n=20 proximal lesions and n=54 occlusal lesions.  Lesions were selected based on their presence on microCT images and visibility in at least one of the SWIR images.  Lesions were chosen that had penetrated at least half-way through the enamel.  Large cavitated lesions or those that penetrated more than half-way through dentin were avoided.

  1. Why were these instruments chosen?

Response: The rationale for using SWIR occlusal transillumination and reflectance (SWIR-OTR) is now better described in the introduction.  SWIR imaging is more sensitive than radiographs and does not require ionizing radiation.  The main reason to use SWIR-OTR is to increase diagnostic performance by reducing the potential for false positives.  Secondary reasons are to gain additional information about lesion severity and increase visibility of tooth surfaces.  This study seeks to determine the highest lesion contrast for transillumination and reflectance at the SWIR wavelengths to determine the best wavelengths to use for SWIR-OTR. 

  1. The statistical analysis paragraph is missing

Response: We apologize for leaving out the statistical approach.  This has been added to methods section. Repeated Measures ANOVA with Tukey’s multiple comparison test was used to compare the lesion contrast across wavelengths for each lesion type and contrast calculation method.  Prism statistical software from GraphPad Software, Inc., (La Jolla, CA) was used for the calculations. Significance level was set at P < 0.05. The statistical results are presented in Table 1.  Means with the same letter across each row are statistically similar (P>0.05).   

  1. LINE 247: The x-ray should be presented in the image.

Response:  The radiograph of the tooth has been added to Figure 4.

  1. Authors should state what are the limits of their study and where they think they could have improved their research.

Response:  We have added to the discussion: This in vitro study lacks the presence of adjacent teeth and the conditions of the oral environment that would be encountered for clinical imaging.   Those factors more profoundly influence diagnostic performance comparisons with radiography and should be less important for the lesion contrast comparisons between the SWIR wavelengths that were carried out in this study.

Round 2

Reviewer 1 Report

Comments and Suggestions for Authors

Dear author, I think you have forgotten an issue that I asked you to add in my previous review. Effect size tells us how meaningful the relationship between variables or the difference between groups is. It indicates the practical significance of a research outcome. Please indicate the effect size and power of your study. A large effect size means that a research finding has practical significance, while a small effect size indicates limited practical applications. This will help us better understand the scope of your study results. I would also appreciate it if you could share the reference article you used to calculate the sample size in your answer.

Author Response

Thank you very much for taking the time to review this manuscript. We appreciate your efforts to improve the manuscript.   

Reviewer#1

Comments: Dear author, I think you have forgotten an issue that I asked you to add in my previous review. Effect size tells us how meaningful the relationship between variables or the difference between groups is. It indicates the practical significance of a research outcome. Please indicate the effect size and power of your study. A large effect size means that a research finding has practical significance, while a small effect size indicates limited practical applications. This will help us better understand the scope of your study results. I would also appreciate it if you could share the reference article you used to calculate the sample size in your answer.

Response:

Based on the first row of Table 1, the proximal lesions the contrast (conventional calculation) had a high of 0.57 and a low of 0.20.  Contrast varies from 0 to 1 and 0.57 is a very high contrast.  Using the mean(s.d) for these two values 0.57(0.14) and 0.20(0.20) and n=20, Cohen’s d is 2.14 which is a very large effect size.

An effect size of 0.65 and sample size of 20 with alpha =0.05 would yield a power of 0.95. 

In Ng et al. reference 23 the contrast was measured at several wavelengths (n=55) for the contrast high, the mean(s.d) was 0.68(0.21) and the lowest positive contrast was 0.066(0.23) which yields a Cohen’s d of 2.8.

The goal of this study was to determine the optimal wavelengths to be used for SWIR reflectance and transillumination imaging and the wavelengths that yield significantly higher contrast are best.  There are very large differences in the lesion contrast from 1000-1750 nm and those differences are expected to impact the diagnostic performance. 

Reviewer 2 Report

Comments and Suggestions for Authors

Dear Authors,

Thank You for Your corrections.

Sincerely, Reviewer

Author Response

Thank you very much for taking the time to review this manuscript. We appreciate your efforts to improve the manuscript.   

Reviewer 2

Comments: Dear Authors,

Thank You for Your corrections.

Sincerely, Reviewer

Response: We appreciate that are response has been sufficient.